# Single-color, ratiometric biosensors for detecting signaling activities in live cells

Brian L Ross[1,2], Brian Tenner[1,3], Michele L Markwardt[4], Adam Zviman[4], Guoli Shi[5], Jaclyn P Kerr[5], Nicole E Snell[4], Jennifer J McFarland[4], Joseph R Mauban[4], Christopher W Ward[5], Megan A Rizzo[4]*, Jin Zhang[1,3,6]*

[1]Department of Pharmacology, University of California, San Diego, San Diego, United States; [2]Department of Biomedical Engineering, Johns Hopkins University, Baltimore, United States; [3]Program in Molecular Biophysics, Johns Hopkins University School of Medicine, Baltimore, United States; [4]Department of Physiology, University of Maryland Baltimore, Baltimore, United States; [5]Department of Orthopaedics, University of Maryland Baltimore, Baltimore, United States; [6]Department of Pharmacology and Molecular Sciences, Johns Hopkins University School of Medicine, Baltimore, United States

**Abstract** Genetically encoded fluorescent biosensors have revolutionized the study of signal transduction by enabling the real-time tracking of signaling activities in live cells. Investigating the interaction between signaling networks has become increasingly important to understanding complex cellular phenomena, necessitating an update of the biosensor toolkit to allow monitoring and perturbing multiple activities simultaneously in the same cell. We therefore developed a new class of fluorescent biosensors based on homo-FRET, deemed FLuorescence Anisotropy REporters (FLAREs), which combine the multiplexing ability of single-color sensors with a quantitative, ratiometric readout. Using an array of color variants, we were able to demonstrate multiplexed imaging of three activity reporters simultaneously in the same cell. We further demonstrate the compatibility of FLAREs for use with optogenetic tools as well as intravital two-photon imaging.

*For correspondence:
mrizz001@umaryland.edu (MAR);
jzhang32@ucsd.edu (JZ)

Competing interests: The authors declare that no competing interests exist.

## Introduction

Genetically encoded biosensors have revolutionized the study of cell signaling by allowing the real-time monitoring of signaling activities, such as enzymatic activity or the release of second messengers, in live cells. They are therefore critical tools for uncovering the precise spatial and temporal regulation of signal transduction cascades. These biosensors can be divided into two broad classes: single-color and ratiometric. Single-color sensors, with an intensiometric activity readout, only occupy a single-color channel, allowing for more flexibility in multiplexed imaging experiments. However, they are sensitive to variations in probe concentration caused by changing expression levels or cell shape, as well as differences in imaging conditions, such as illumination intensity and focus. On the other hand, sensors with a ratiometric readout, such as those based on Förster Resonance Energy Transfer (FRET), cancel out many of these variations, enabling quantitative measurements of second messenger concentrations and better comparisons between experiments. However, the requirement for two distinct color channels limits their application in multiplexed imaging.

We therefore aimed to develop sensors that only occupy a single channel while still cancelling out the effects of varying imaging conditions and probe concentrations. Hence, rather than using the emission ratio between the FRET and donor channels, we instead used the loss of polarization of emitted light as a readout for FRET between two fluorescent proteins (FPs). Because this approach does not require the donor and acceptor to have distinct emission wavelengths, it can be used for

either hetero-FRET (e.g. between different chromophores) or homo-FRET (e.g. between identical chromophores). Homo-FRET measurements have been useful for detecting protein clustering and protein oligomerization in live cells (*Bader et al., 2011*, *Bader et al., 2009*; *Gautier et al., 2001*), but only recently has the possibility of using homo-FRET in biosensor designs been explored (*Warren et al., 2015*; *Cameron et al., 2016*). Here, we describe the development of a panel of single-color, genetically encodable biosensors based on homo-FRET for detecting kinase activity and second messenger dynamics. We call these sensors FLuorescence Anisotropy REporters, or FLAREs.

## Results and discussion

To create our FLARE probes, we adapted existing FRET-based biosensors for homo-FRET measurements by replacing the traditional FRET pair with two FPs of the same color. The resulting biosensors include a molecular switch, which changes conformation in the presence of a particular biochemical activity, flanked by two spectrally similar FPs at the N- and C-termini. Changes in the conformation of the molecular switch, and thus the biochemical activities under study, are then monitored by observing the fluorescence anisotropy of the sensor using fluorescence polarization microscopy, with increased anisotropy corresponding to a lower-FRET state of the sensor, similar to the effect of increasing the intramolecular distance between the FRET pair (*Figure 1—figure supplement 1*).

To develop a Protein Kinase A (PKA) activity FLARE, the molecular switch from A Kinase Activity Reporter 4 (AKAR4) (*Zhang et al., 2001*; *Depry et al., 2011*), composed of an FHA1 domain and PKA substrate (*Figure 1a*), was flanked between two FPs of the same color. The FHA1 domain binds to the PKA substrate when the latter is phosphorylated, altering the conformation of the molecular switch and leading to a change in FRET between the flanking homo-FRET pair. To test the effect of FP circular permutation on these biosensors, we developed two FLARE-AKAR variants based on the yellow FP mVenus: one in which the C-terminal FP was circularly permuted at position 172 (cp172Venus), consistent with the hetero-FRET AKAR4 sensor, and one without circular permutation. We expressed mVenus-cp172Venus FLARE-AKAR in HEK293T cells and captured a time-course using fluorescence polarization microscopy. Following PKA activation using a cocktail of 50 μM forskolin (Fsk), an adenylyl cyclase activator, and 100 μM 3-isobutyl-1-methylxanthine (IBMX), a general phosphodiesterase inhibitor, the anisotropy decreased from $0.29 \pm 0.003$ to $0.26 \pm 0.003$, a decrease of $0.028 \pm 0.001$ (N = 44, biological replicates, unpaired, two-tailed t-test, p<0.0001), with the kinetics of the decrease matching those observed with AKAR4 (*Figure 1b*, *Figure 1—figure supplement 2a*). The yellow variant without the circular-permutation in the C-terminal FP showed slightly reduced changes in anisotropy upon stimulation with Fsk/IBMX (*Figure 1c*, *Figure 1—figure supplement 2c*), consistent with previous

observations in hetero-FRET-based biosensors (*Nagai et al., 2004*; *DiPilato and Zhang, 2009*; *Allen and Zhang, 2006*). We observed a slight positive correlation between intensity and anisotropy change for Venus-cp172Venus FLARE AKAR; however, the expression level does not significantly impact the reporting ability of these sensors in general (*Figure 1—figure supplement 3a*). The signal-to-noise ratio (SNR) of Venus-cp172Venus FLARE-AKAR was calculated to be 32 by dividing the magnitude of the anisotropy change upon maximal PKA stimulation by the standard deviation of the baseline before stimulation.

Subsequent control experiments confirmed that this change in anisotropy is caused by a change in the FRET state due to the conformational change of the sensor upon stimulation of PKA activity. PKA inhibition using 20 μM H-89 led to an immediate slope change and increase in anisotropy (*Figure 1b*). On the other hand, a mutant version of the biosensor with a threonine-to-alanine (T-to-A) mutation at the phosphorylation site showed no change in anisotropy upon PKA stimulation with Fsk/IBMX or inhibition with H-89 (*Figure 1b*, *Figure 1—figure supplement 2b*), suggesting that the observed changes in anisotropy were due to phosphorylation of the PKA substrate. Furthermore, isoproterenol, a β-adrenergic agonist, induced FLARE-AKAR responses in a dose-dependent manner (*Figure 1—figure supplement 4*). To further demonstrate that the change in anisotropy upon PKA stimulation was due to a change in FRET, we mutated the chromophore of the C-terminal cp172Venus in Venus-cp172Venus FLARE AKAR from GYG to GGG. We observed that the magnitude of the response to Fsk/IBMX decreased to approximately one-third of that of the wild-type sensor (*Figure 1—figure supplement 5*). The remaining response was likely due to intermolecular FRET that

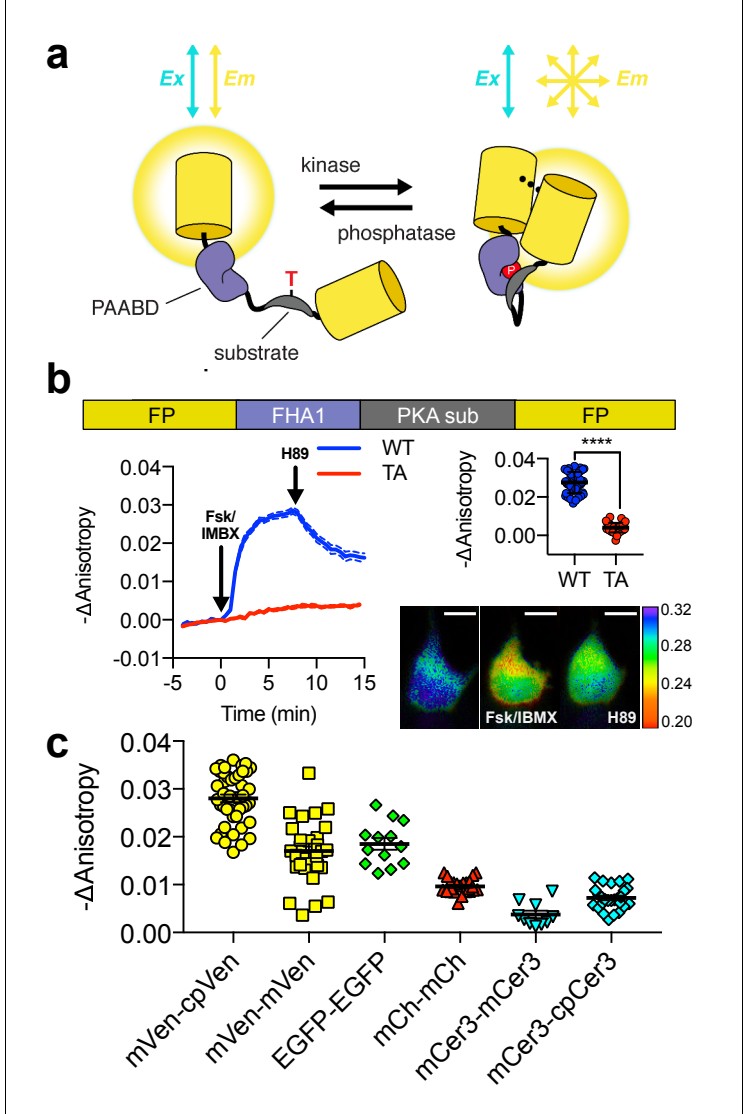

**Figure 1.** Design and characterization of FLARE AKAR. (**a**) Schematic of a kinase activity FLARE (**b**) Diagram illustrating domain structure of FLARE-AKAR (top). Time-course of mean fluorescence anisotropy of Venus-cp172Venus FLARE-AKAR wild type (blue, N = 44) and kinase insensitive T-to-A mutant (red, N = 38) expressed in HEK293T cell, stimulated with 50 µM forskolin and 100 µM IBMX at t = 0 min, and 20 µM H-89 at t = 24 min (left). Dashed lines above and below represent standard error of the mean. Changes in anisotropy upon Fsk/IBMX stimulation for both FLARE-AKAR WT and T-to-A mutant (upper right, two-tailed t-test, p<0.0001), calculated as the difference between the mean anisotropy from t = 5 min to t = 7.5 min and the mean anisotropy of the baseline before drug addition. The mean for each is shown, with the error reflecting the standard error of the mean. Representative anisotropy pseudocolor image before Fsk/IBMX stimulation (t = 0 min), after Fsk/IBMX stimulation (t = 7.5 min), and after inhibition of PKA with H-89 (t = 24 min) (lower right) (**c**) Comparison of the magnitude of the anisotropy change for different color variants of FLARE-AKAR upon stimulation with Fsk/IBMX including Venus-cp172Venus AKAR (N = 44), Venus-Venus FLARE AKAR (N = 32), EGFP-EGFP FLARE AKAR (N = 13), mCh-mCh FLARE AKAR (N = 22), mCerulean3 FLARE-AKAR (N = 10), and mCerulean3-cp173Cerulean FLARE-AKAR (N = 26). The mean for each is shown, with the error reflecting the standard error of the mean. The online version of this article includes the following source data and figure supplement(s) for figure 1:

**Source data 1.** FLARE AKAR characterization.
**Figure supplement 1.** Anisotropy vs. intramolecular distance.
**Figure supplement 2.** Individual and average traces for FLARE-AKAR panel.
**Figure supplement 3.** Anisotropy change vs expression level.
**Figure supplement 4.** Venus-cp172Venus FLARE-AKAR isoproterenol dose response.

*Figure 1 continued*

**Figure supplement 5.** Characterization of the chromophore-dead FLARE AKAR mutant.
**Figure supplement 6.** Direct comparison of mCherry-mCherry FLARE-AKAR and heteroFRET AKAR4.
**Figure supplement 7.** Subcellular targeted FLARE-AKARs.
**Figure supplement 8.** Differential PKA activity kinetics in the cytosol and the nucleus.

occurs when the FHA1 domain of one molecule binds to the phosphorylated PKA substrate in an adjacent molecule.

In addition to the yellow sensors, we developed a panel of color variants of FLARE-AKAR, including EGFP-EGFP, mCherry-mCherry, mCerulean3-mCerulean3 (*Markwardt et al., 2011*), and mCerulean3-cp173Cerulean3 versions. All these variants exhibited a decrease in anisotropy in cells treated with Fsk/IBMX; however, the magnitude of the anisotropy decrease depended on the choice of FP, with the mVenus-cp172Venus variant having the largest response (*Figure 1c*, *Figure 1—figure supplement 2*). As with the Venus variants, the Cerulean3-based FLARE-AKAR showed an increased dynamic range with a circularly permutated fluorescent protein at the C-terminal position. mCherry-mCherry FLARE-AKAR, being spectrally shifted from the AKAR4 heteroFRET sensor allowed for direct comparison of FLARE and heteroFRET sensors within the same cell. As shown in *Figure 1—figure supplement 6*, changes in anisotropy in mCherry-mCherry FLARE-AKAR corresponded with the changes in normalized emission ratio in AKAR4, with similar kinetics.

We furthermore demonstrated the ability of FLARE-AKAR sensors to monitor kinase activity at particular subcellular compartments. By fusing Venus-cp172Venus FLARE-AKAR to targeting motifs from Lyn kinase and DAKAP1, we were able to detect PKA activity at the plasma membrane and outer mitochondrial membrane, respectively (*Figure 1—figure supplement 7*). Moreover, we used untargeted FLARE sensors to detect differential PKA activity dynamics in different compartments; diffusable Venus-cp172Venus FLARE AKAR in HeLa showed that PKA activity has slower kinetics and a lower magnitude in the nucleus than the cytosol (*Figure 1—figure supplement 8*).

To demonstrate the generalizability of FLAREs, we developed a family of single-color kinase activity or activation reporters in various colors (*Figure 2*). To construct a single-color Erk activity biosensor, we replaced the PKA sensor domain from Venus-cp172Venus FLARE-AKAR with the sensor domain from EKAR-EV, composed of a WW domain (PAABD), a flexible EV linker, and an Erk substrate peptide (*Figure 2a*) (*Harvey et al., 2008*; *Vandame et al., 2014*). When expressed in HEK293T cells, Venus-cp172Venus FLARE-EKAR-EV exhibited a decrease in anisotropy of $0.02 \pm 0.001$ (N = 13) after treatment with 100 ng/mL epidermal growth factor (EGF) to activate the MAPK pathway (*Figure 2a*, *Figure 2—figure supplement 1*). This response was reversed upon MEK inhibition using 20 μM U0126, and no change in anisotropy was observed with a T-to-A mutant sensor. Likewise, we developed a panel of PKC activity reporters, called FLARE-CKARs, by flanking a PKC sensor domain composed of an FHA1 domain and a PKC substrate (*Herbst et al., 2011*) from a CKAR2 construct (*Figure 2—figure supplement 2*) with mVenus-cp172Venus. We observed an anisotropy decrease of $0.02 \pm 0.001$ (N = 26) upon activation of PKC with 100 ng/mL phorbol 12-myristate 13-acetate (PMA) (*Figure 2b*, *Figure 2—figure supplement 3*). For both FLARE-EKAR (*Figure 2—figure supplement 1*) and FLARE CKAR (*Figure 2—figure supplement 3*), we likewise repeated this process for various color variants, and like the FLARE-AKARs, the sensors based on mVenus exhibited the largest responses. Furthermore, a myosin light chain kinase (MLCK) sensor was converted to a FLARE by exchanging the cyan FP for mVenus (*Isotani et al., 2004*). Calmodulin (CaM) association with the MLCK-CaM binding domain in between the FPs decreases FRET, leading to an increase in fluorescence anisotropy upon forced calcium ($Ca^{2+}$) entry with 30 mM KCl (N = 13) (*Figure 2c*, *Figure 2—figure supplement 4*).

In addition to biosensors for monitoring enzymes, we also developed FLAREs for monitoring second messenger dynamics (*Figure 3a*). We developed a $Ca^{2+}$ FLARE by utilizing the sensor domain from the Cameleon family of biosensors, composed of CaM and the $Ca^{2+}$/CaM-binding peptide M13 (*Nagai et al., 2004*; *Miyawaki et al., 1999*). When expressed in HEK293T cells, Venus-cp172Venus FLARE-Cameleon exhibited a decrease in anisotropy of $0.03 \pm 0.002$ (N = 10) upon addition of 1 μM ionomycin and 5 mM $CaCl_2$, with the Venus-Venus, mCerulean3-mCerulean3, and

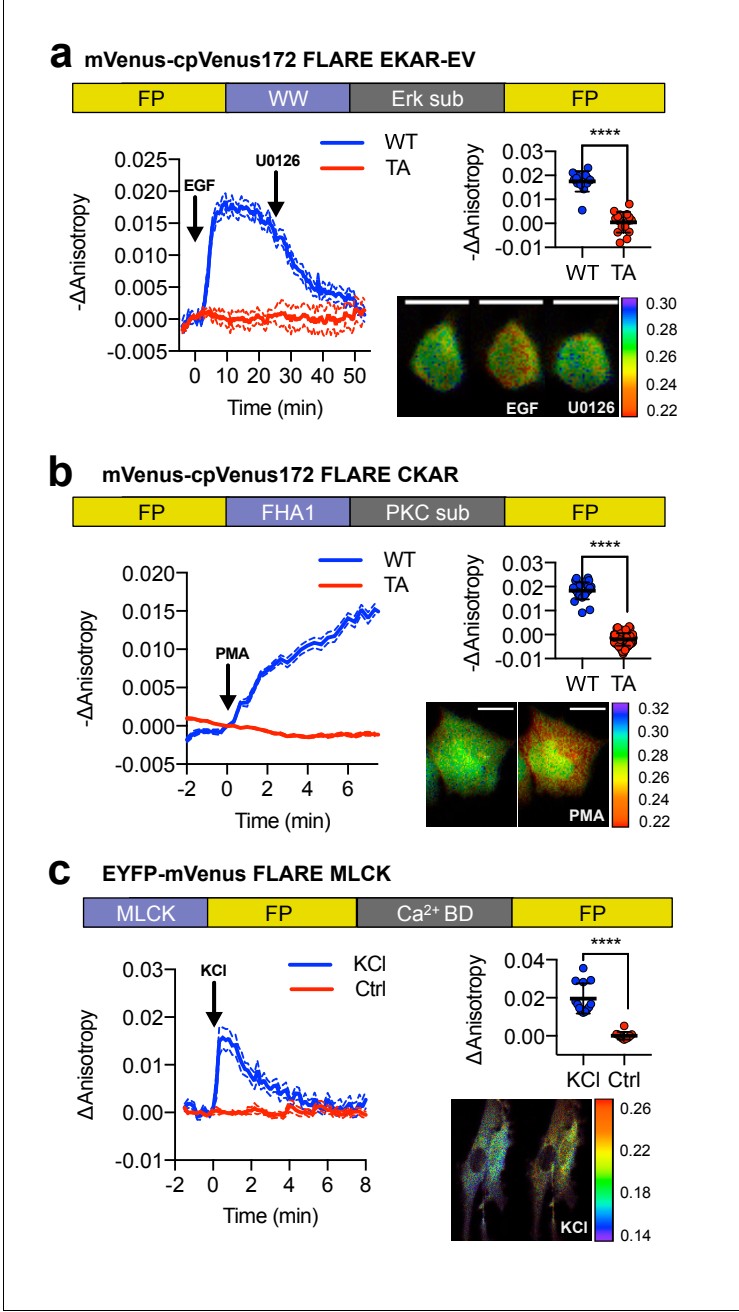

**Figure 2.** A panel of kinase activity and activation biosensors. (**a**) Domain structure of FLARE-EKAR-EV (above). Time-course of mean fluorescence anisotropy of Venus-cp172Venus FLARE-EKAR-EV WT (blue, N = 13) and kinase-insensitive mutant (red, N = 16) expressed in HEK293T cell, with addition of 100 ng/mL EGF at t = 0 min and 20 μM U0126 at t = 25 min (left). Summary of anisotropy changes (upper right, two-tailed t-test, p<0.0001), calculated as the difference between the mean anisotropy from t = 10 min to t = 15 min and the mean anisotropy of the baseline before drug addition. The mean is shown, with the error reflecting the standard error of the mean. Representative anisotropy pseudocolor image before EGF stimulation (t = 0 min), after EGF stimulation (t = 25 min), and after inhibition of MEK with U0126 (t = 47.5 min) (lower right). (**b**) Domain structure of FLARE-CKAR (above). Time-course of fluorescence anisotropy of Venus-cp172Venus FLARE-CKAR WT (blue, N = 26) and kinase-insensitive mutant (red, N = 119) with addition of 100 ng/mL phorbol 12-myristate 13-acetate (PMA) at t = 0 min. Summary of response magnitudes (upper right, two-tailed t-test, p<0.0001), calculated as the difference between the mean anisotropy from t = 10 to t = 11.33 min and the mean anisotropy of the baseline before drug addition. The mean is shown, with the error reflecting the standard error of the mean. Representative anisotropy pseudocolor image before PMA addition (t = 0 min) and after PMA addition (t = 15 min). (**c**) Domain structure of

*Figure 2 continued on next page*

*Figure 2 continued*

FLARE-MLCK (top). Anisotropy time course of a representative REF52 cell expressing YFP-Venus FLARE-MLCK treated with 30 mM KCl at t = 0 (N = 13, blue), or vehicle control (N = 10). Summary of anisotropy changes, calculated as the mean difference between the anisotropy at t = 0.333 min and the anisotropy of the baseline, before KCl addition (upper right, two-tailed t-test, p<0.0001). Representative pseudocolor anisotropy images before and after KCl treatment (lower right). Dashed lines above and below time course traces reflect the standard error of the mean. FP, fluorescent protein; CaM BD, MLCK calmodulin-binding domain.

The online version of this article includes the following source data and figure supplement(s) for figure 2:

**Source data 1.** FLARE kinase biosensor panel.
**Figure supplement 1.** FLARE-EKAR characterization.
**Figure supplement 2.** Characterization of the CKAR2 hetero-FRET biosensor.
**Figure supplement 3.** FLARE CKAR characterization.
**Figure supplement 4.** FLARE MLCK characterization.

mCherry-mCherry versions also showing detectable responses (*Figure 3b*, *Figure 3—figure supplement 1a,b*). The mVenus-based FLARE sensors tend to show larger dynamic ranges for a variety of FLARE sensors than mCerulean3 or mCherry variants, likely due to the superior extinction coefficient and quantum yield of mVenus, which make it a good FRET donor and acceptor. We further demonstrated the ability of Venus-cp172Venus FLARE Cameleon to detect submaximal responses by monitoring calcium transients in histamine-stimulated HeLa cells (*Figure 3—figure supplement 1c*). In order to determine the dissociation constant and Hill coefficients, we purified Venus-cp172Venus Cameleon and measured the fluorescence anisotropy in solutions of known free $Ca^{2+}$ concentration at different temperatures (*Figure 3—figure supplement 2*). The resulting parameters are in good agreement with other Cameleon sensors (*Nagai et al., 2004*). Furthermore, we developed another calcium FLARE sensor, based on D1-ER (*Palmer et al., 2004*), with a sensitivity appropriate for calcium monitoring in the ER. The anisotropy decreases upon increasing calcium concentration and increases when calcium is depleted from the ER upon treatment with thapsigargin (*Figure 3—figure supplement 3*). In addition to $Ca^{2+}$, we developed a FLARE to detect intracellular cAMP based on the ICUE family of sensors, in which a conformational change in a truncated form of the cAMP effector Epac leads to a decrease in FRET efficiency in the presence of cAMP (*DiPilato and Zhang, 2009*) (*Figure 3a*). When expressed in HEK293T cells, the fluorescence anisotropy of Venus-cp172Venus FLARE-ICUE increased by 0.02 ± 0.001 (N = 40) upon stimulation with Fsk/IBMX (*Figure 3c*, *Figure 3—figure supplement 4*).

The fact that FLAREs only occupy a single-color channel and are highly generalizable for different biosensors, as well as color variants, highlights their utility for multiplexed imaging applications. While multiplexing of hetero-FRET sensors is generally limited to two probes (*Depry et al., 2013*; *Shcherbakova et al., 2012*), we demonstrate that FLAREs can facilitate co-imaging of three biosensors simultaneously. We co-expressed mCherry-mCherry FLARE-AKAR, Venus-cpVenus FLARE-EKAR-EV, and mCerulean3-mCerulean3 FLARE-Cameleon in HEK293T cells and acquired a time-course with sequential treatment using Fsk/IBMX, EGF, and thapsigargin. Clear and distinct decreases in anisotropy were observed in the red channel after Fsk/IBMX treatment, in the yellow channel after EGF stimulation, and in the cyan channel after thapsigargin treatment, corresponding to an increase in PKA activity, Erk activity, and intracellular $Ca^{2+}$, respectively (N = 17) (*Figure 4a*, *Figure 4—figure supplement 1*).

We further aimed to show that FLARE sensors can be used to monitor multiple signaling activities simultaneously in different cellular contexts. For example, mCer3-mCer3 FLARE Cameleon, co-imaged with other FLARE sensors in HeLa cells, can detect calcium responses to physiologically relevant stimulation conditions, such as histamine (*Figure 4—figure supplement 2*). Additionally, when mCer3-mCer3 FLARE AKAR was co-imaged with Venus-cp172Venus cameleon in HEK293T cells, activation of the β-adrenergic with isoproterenol led to a transient decrease in anisotropy in the cyan-channel (*Figure 4—figure supplement 3*). Furthermore, we used FLAREs to study the cAMP-$Ca^{2+}$ oscillatory circuit in pancreatic β-cells. MIN6 β-cells were transiently transfected with Venus-cp172Venus FLARE-ICUE and mCerulean3-mCerulean3 FLARE-Cameleon to simultaneously monitor cAMP and $Ca^{2+}$ dynamics, respectively (N = 19). We observed clear fluorescence anisotropy oscillations in both channels following stimulation with 20 mM tetraethylammonium chloride (TEA)

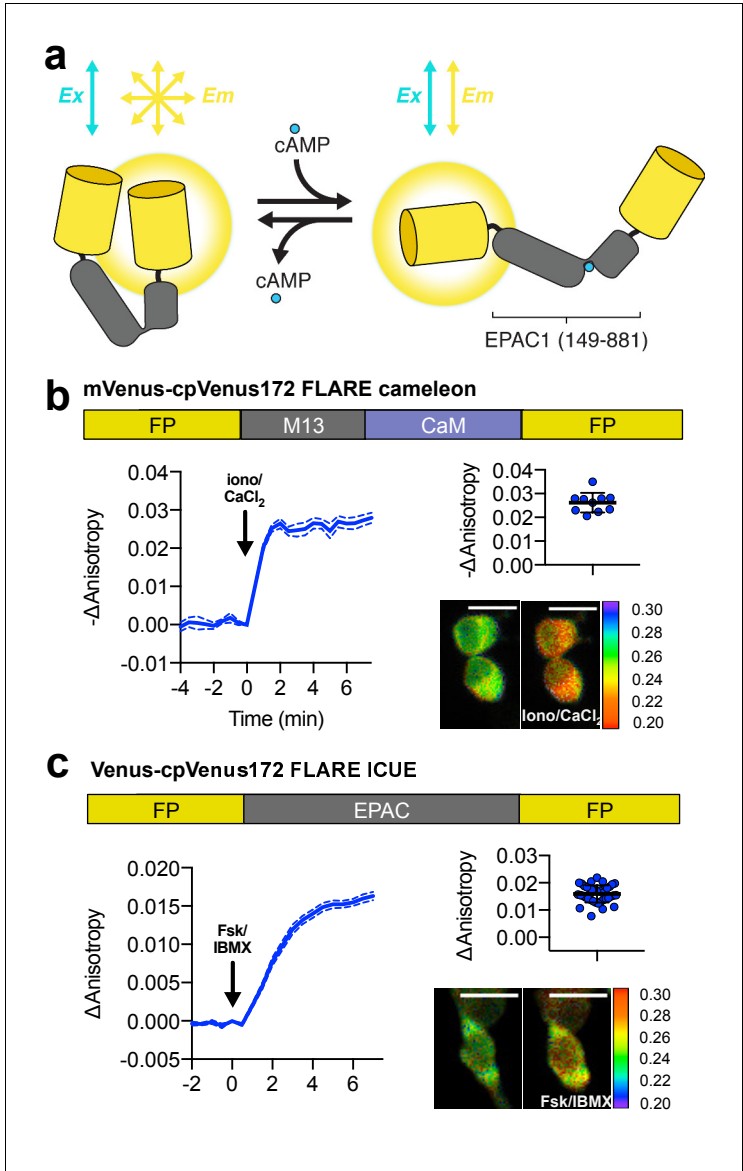

**Figure 3.** Design and characterization of FLARE second messenger biosensors. (**a**) Schematic of FLARE ICUE cAMP biosensor. (**b**) Domain structure of FLARE-Cameleon (top). Time-course of mean fluorescence anisotropy of Venus-cp172Venus FLARE-Cameleon (N = 10) with addition of 1 μM ionomycin and 5 mM $CaCl_2$. Summary of anisotropy changes after 1 μM ionomycin and 5 mM $CaCl_2$ (upper right) calculated as the difference between the mean anisotropy from t = 5 min to t = 7.5 min and the anisotropy of the baseline before drug addition Representative anisotropy pseudocolor image before and after 1 μM ionomycin and 5 mM $CaCl_2$ addition. (**c**) Domain structure of the cAMP biosensor FLARE-ICUE (top). Time-course of mean fluorescence anisotropy of Venus-cp172Venus FLARE-ICUE (N = 40) with addition of 50 μM forskolin and 100 μM IBMX at t = 0 (left). Summary of anisotropy changes after 50 μM forskolin and 100 μM IBMX with respect to baseline (upper right), calculated as the difference between the mean anisotropy from t = 5 min to t = 7.5 min and the anisotropy of the baseline before drug addition. Representative anisotropy pseudocolor image before (t = 0 min) and after (t = 7.5) stimulation with Fsk/IBMX (lower right). Dashed lines above and below time course reflect standard error of the mean.

The online version of this article includes the following source data and figure supplement(s) for figure 3:

**Source data 1.** FLARE second messenger biosensor panel.
**Figure supplement 1.** Characterization of FLARE Cameleon.
**Figure supplement 2.** In vitro calibration of purified Venus-cp172 FLARE-Cameleon.
**Figure supplement 3.** Characterization of CFP FLARE-D1ER.
**Figure supplement 4.** Venus-cp172Venus FLARE ICUE single cell traces.

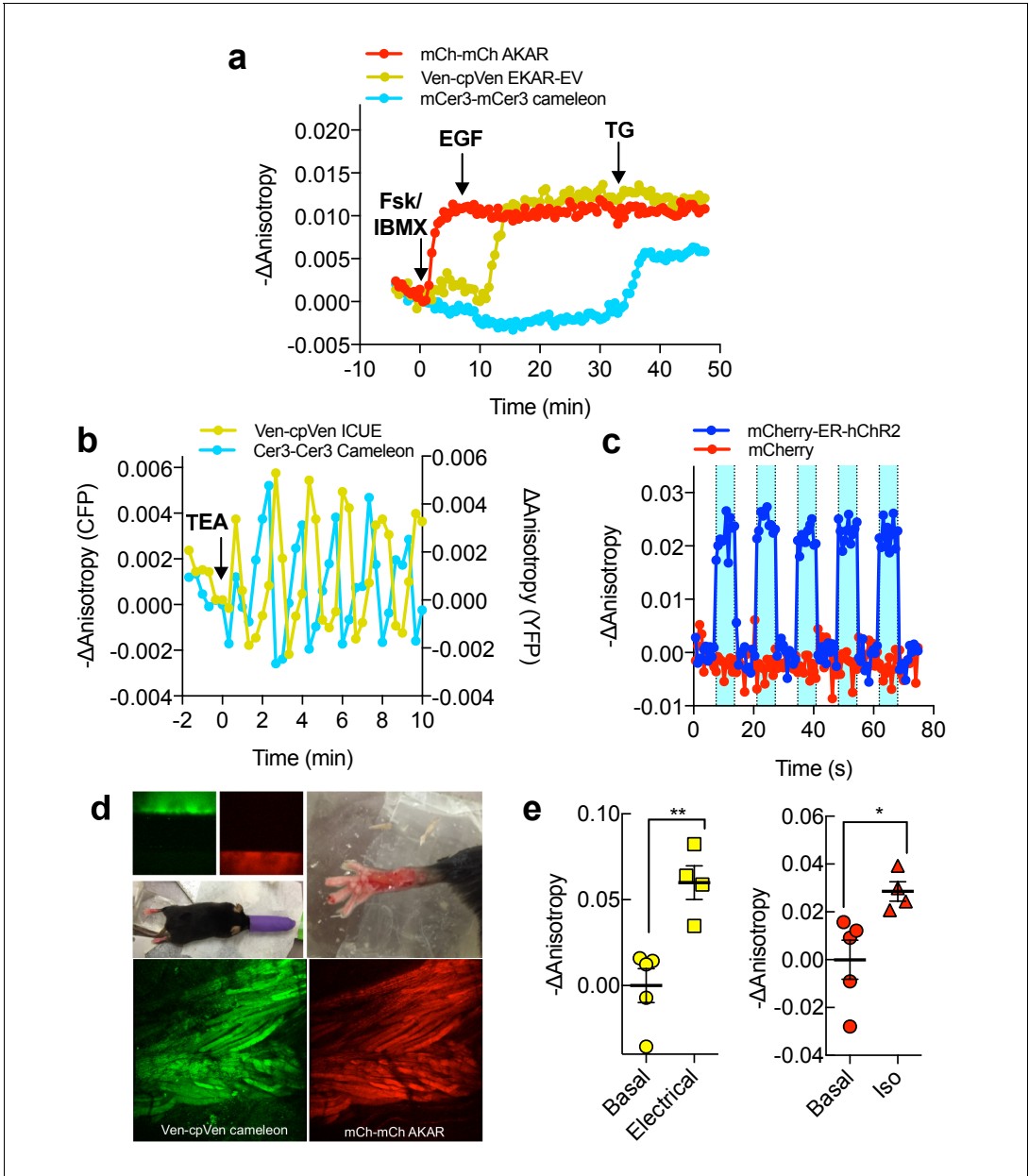

**Figure 4.** Multiparameter Imaging with FLAREs. (a) Time-course of a representative HEK293T cell co-expressing mCherry-mCherry FLARE-AKAR, mVenus-cp172Venus FLARE-EKAR-EV, and mCerulean3-mCerulean3 FLARE-Cameleon, with 50 μM forskolin and 100 μM IBMX added at t = 0 min, 100 ng/mL EGF at t = 7.5 min, and 1 μM thapsigargin added at t = 32.5 min (N = 17). (b) Anisotropy response of a representative MIN6 cell co-expressing Cerulean3-Cerulean3 FLARE-Cameleon and Venus-cp172Venus FLARE-ICUE, showing robust oscillations after stimulation with TEA at t = 0 min (N = 19). (c) Venus-cp172Venus FLARE-Cameleon anisotropy time-course in REF52 cells co-expressing the sensor and either mCherry-tagged humanized channel rhodopsin2 (blue-curve) or mCherry alone (red-curve), with periodic illumination with 455 nm light (indicated by cyan shading). (d) Purified mVenus and mCherry proteins were placed in separate capillary tubes and imaged with 855 nm and 1200 nm two-photon excitation. Plasmids encoding Venus-cp172Venus FLARE-Cameleon and mCherry-mCherry FLARE-AKAR were electroporated into the skeletal muscle of the foot of a live mouse for in vivo imaging. Below are Z-stack projections of skeletal muscle co-expressing these two sensors, excited with 855 nm (left) and 1200 nm (right) light. (e) Electrical stimulation decreased fluorescence anisotropy in the 855 nm channel, consistent with increased cytoplasmic $Ca^{2+}$ (left) (two-tailed T-test, p=0.0037). Intraperitoneal injection of isoproterenol activates mCherry-mCherry FLARE AKAR (two-tailed T-test, p=0.0239), as indicated by decreased anisotropy in the 1200 nm channel (right). The mean for each is shown, with the error reflecting the standard error of the mean.

The online version of this article includes the following source data and figure supplement(s) for figure 4:

**Source data 1.** Multiparameter imaging of FLAREs.
**Figure supplement 1.** Average and single cell traces for multiplexed imaging of PKA activity, Erk activity, and calcium in HEK293-T cells.
**Figure supplement 2.** Multiplexed monitoring of PKA activity, Erk activity, and calcium in HeLa cells, with histamine stimulation.
*Figure 4 continued on next page*

*Figure 4 continued*

**Figure supplement 3.** Simultaneous monitoring of PKA and calcium in live cells with FLAREs using isoproterenol stimulation.

**Figure supplement 4.** Monitoring calcium and cAMP oscillations in pancreatic β cells.

**Figure supplement 5.** Individual (gray) and average (red) cell traces for Venus-cp172Venus FLARE AKAR co-expressed with hChR2-ER, with intermittent exposures to 455 nm light (N = 11).

(*Figure 4b*, *Figure 4—figure supplement 4*). $Ca^{2+}$ and cAMP exhibit synchronized oscillations, with $Ca^{2+}$ increases corresponding to cAMP decreases, consistent with previous findings (*Landa et al., 2005*; *Ni et al., 2011*). These data demonstrate that even the lower signal mCerulean3-based FLARE sensors can be used under sub-maximal and physiologically relevant stimulation conditions in multiplexed imaging experiments.

In addition to multiplexed biosensor imaging, the fact that FLARE sensors occupy only a single color channel also permits simultaneously perturbing and monitoring biochemical activities using optogenetics and FLAREs, respectively. We coexpressed an mCherry-tagged, ER-targeted channelrhodopsin2 (hChR2) (*Nagel et al., 2003*; *Lin et al., 2009*; *Markwardt et al., 2016*), a light-gated calcium ion channel, with Venus-cp172 FLARE-Cameleon in REF52 cells, a rat embryonic fibroblast cell line. Illumination with blue light produced an immediate decrease in anisotropy, corresponding to an increase in intracellular $Ca^{2+}$ (*Figure 4c*, *Figure 4—figure supplement 5*). Control cells lacking hChR2 expression showed no change in anisotropy in the yellow channel.

In vivo, two-photon imaging of FLAREs was tested using a skeletal muscle preparation (*Figure 4d,e*). Exclusive excitation of mVenus (855 nm) or mCherry (1200 nm) was verified by imaging capillary tubes filled with recombinant proteins (*Figure 4d*). Plasmids encoding Venus-cp172Venus FLARE-Cameleon and mCherry FLARE-AKAR sensors were electroporated into the flexor digitorum brevis muscle of a live mouse (*DiFranco et al., 2009*; *Kerr et al., 2015*). Administration of an electrical current stimulated a rise in intracellular $Ca^{2+}$ concentration, as indicated by a decrease in FLARE-Cameleon anisotropy, independent of changes in FLARE-AKAR anisotropy (*Figure 4e*). Activation of AKAR was then induced by intraperitoneal injection of isoproterenol (0.5 mg/kg). Thus, FLAREs enable in vivo multiparametric biosensor measurements.

We have demonstrated that FLAREs are a highly generalizable, accessible platform for creating single-color sensors to detect biochemical activities in individual cells in real time. Their ratiometric readout allows for fluctuations in light intensity and probe concentration to be cancelled out, permitting quantitative measurements of intracellular concentrations. We showed that these sensors have an SNR of 3–32 (*Table 1*) and a dynamic range comparable with first-generation FRET sensors (*Newman and Zhang, 2008*; *Zhou et al., 2015*). Future development and optimization should further enhance their performance. We demonstrated that current FLAREs are already useful for multiplexed imaging applications. They can also be used in conjunction with optogenetic tools to enable all-optical interrogation of cellular regulation, and for intravital two-photon imaging to facilitate studies in tissues and living animals. FLAREs, by allowing researchers to monitor multiple activities within the same cell, as well as to both monitor and optogenetically perturb activities in the same cell, could be used to study the how the spatiotemporal regulation of biochemical activities in highly integrated pathways allows a small number of signals to produce diverse cellular behaviors.

## Materials and methods

**Key resources table**

| Reagent type (species) or resource | Designation | Source or reference | Identifiers |
|---|---|---|---|
| Cell Line (*Homo sapiens*) | HEK293-T | | RRID:CVCL_0063 |
| Cell Line (*Homo sapiens*) | HeLa | | RRID:CVCL_0030 |
| Cell Line (*Mus musculus*) | Min6 | | RRID:CVCL_0431 |

*Continued on next page*

*Continued*

| Reagent type (species) or resource | Designation | Source or reference | Identifiers |
|---|---|---|---|
| Cell Line (*Rattus norvegicus*) | REF-52 | | RRID:CVCL_6848 |
| Chemical Compound, Drug | Forskolin | LC Laboratories | F9929 |
| Chemical Compound, Drug | 3-Isobutyl-1-methylxanthine (IBMX) | Sigma | I5879 |
| Chemical Compound, Drug | H-89 | LC Laboratories | H5239 |
| Chemical Compound, Drug | Epidermal Growth Factor (EGF) | Sigma | E9644 |
| Chemical Compound, Drug | U0126 | Sigma | U120 |
| Chemical Compound, Drug | Phorbol 12-Myristate 13-Acetate (PMA) | LC Laboratories | P-1680 |
| Chemical Compound, Drug | Tetraethylammonium chloride (TEA) | Sigma | T2265 |
| Chemical Compound, Drug | ionomycin | EMD Millipore | 407951 |
| Chemical Compound, Drug | thapsigargin | Cayman Chemical | 10522 |
| Chemical Compound, Drug | histamine | Sigma | H7250 |
| Commerical Kit | Lipofectamine-2000 | Invitrogen | 11668019 |
| Commerical Kit | Polyjet | SignaGen | SL100688 |
| Recombinant DNA Reagent | pCDNA3 AKAR4 | PMID: 20838685 | |
| Recombinant DNA Reagent | pCDNA3 EKAR-EV | PMID: 21976697 | |
| Recombinant DNA Reagent | pCDNA3 CKAR1 | PMID: 12782683 | |
| Recombinant DNA Reagent | FRET MLCK sensor | PMID: 15071183 | |
| Recombinant DNA Reagent | pCDNA3 YC3.6 Cameleon | PMID: 10051607 | |
| Recombinant DNA Reagent | pCDNA3 ICUE3 | PMID: 19603118 | |
| Strain (*Escherichia coli*) | BL-21 Codon Optimized Plus | New England Biolabs | C2527 |
| Software | FIJI | | RRID:SCR_014294 |
| Software | MetaFluor | | RRID:SCR_002285 |
| Software | Micromanager | | RRID:SCR_000415 |
| Software | Zeiss Actiovision | | RRID:SCR_002677 |
| Software | MATLAB | | RRID:SCR_001622 |
| Software | GraphPad Prism | | RRID:SCR_002798 |

## Plasmid and construct construction

Cloning was performed using the pRSET-B vector using typical molecular cloning methods using polymerase chain reaction (PCR) with Phusion polymerase (New England Biolabs), restriction enzyme digestion, and ligation with T4 DNA ligase. To clone Venus-cp172Venus FLARE, AKAR4 was sub-cloned from a modified pCDNA3 to pRSET-B between the BamHI and EcoRI restriction enzyme sites. mVenus was then PCR amplified with primers encoding a BamHI site at the 5' end and an SphI

**Table 1.** Signal to noise ratios for FLARE biosensors.

| Sensor | Signal to noise ratio (± SEM) |
|---|---|
| Venus-cp172Venus FLARE AKAR | 32 ± 2.0 (N = 32) |
| mVenus-mVenus FLARE AKAR | 10 ± 1.6 (N = 32) |
| EGFP-EGFP FLARE AKAR | 6 ± 0.6 (N = 32) |
| mCherry-mCherry FLARE AKAR | 14 ± 1.5 (N = 22) |
| mCerulean3-mCerulean3 FLARE AKAR | 3 ± 0.5 (N = 10) |
| mCerulean3-cp173 Cerulean3 FLARE AKAR | 5 ± 0.3 (N = 26) |
| Venus-cp172Venus FLARE EKAR | 17 ± 3.1 (N = 8) |
| mCherry-mCherry FLARE EKAR | 4 ± 0.8 (N = 10) |
| mCerulean3-mCerulean3 FLARE EKAR | 9 ± 2.3 (N = 9) |
| Venus-cp172Venus FLARE CKAR | 27 ± 3.3 (N = 26) |
| mCherry-mCherry FLARE CKAR | 14 ± 3.0 (N = 8) |
| mCerulean3-mCerulean3 FLARE CKAR | 9 ± 1.7 (N = 6) |
| FLARE-MLCK | 8 ± 0.8 (N = 13) |
| Venus-cp172Venus FLARE Cameleon | 19 ± 4.8 (N = 10) |
| mCherry-mCherry FLARE Cameleon | 5 ± 0.8 (N = 23) |
| mCerulean3-mCerulean3 FLARE Cameleon | 9 ± 1.5 (N = 11) |
| Venus-cp172 Venus FLARE ICUE | 19 ± 1.2 (N = 40) |

site at the 3' end, and the resulting PCR product was digested with BamHI and SphI and ligated to pRSET-B AKAR4 digested with BamHI and SphI, with the mCerulean gene removed. Venus-cp172Venus FLARE AKAR was then sub-cloned back into a modified pCDNA3 vector using the BamHI and EcoRI sites. Other color variants were created by replacing the genes for the fluorescent proteins in other FLARE AKAR variants in pRSET-B, either between the BamHI and SphI sites for the N-terminal fluorescent protein, or SacI and EcoRI for C-terminal fluorescent proteins. Finalized constructs intended for mammalian expression were then sub-cloned into a modified pcDNA3 expression vector between the BamHI and EcoRI sites. FLARE variants of other sensors were created by amplifying the molecular switch from EKAR-EV, CKAR2, Cameleon and ICUE3 with primers encoding the SphI and SacI sites, digesting the PCR product with SphI and SacI enzymes, and ligating them to the relevant pRSET-B FLARE AKAR plasmid digested with SphI and SacI to remove the domains involved in the molecular switch for FLARE AKAR. The final constructs were then subcloned into a modified pCDNA3 expression vector between the BamHI and EcoRI sites. Targeted versions of the sensors were created either by PCR amplifying the sensor with primers containing the targeting sequence and ligating it to the pCDNA3 expression vector between BamHI and EcoRI, or by subcloning the construct into a plasmid already containing the targeting sequence. N-terminal targeting sequences were placed between HindIII and EcoRI, and C-terminal targeting sequences between EcoRI and XbaI. All cloning steps were performed using DH5α strain of *E. coli*.

The threonine to alanine mutants for Venus-cp172Venus FLARE AKAR and FLARE EKAR were created by performing site-directed mutagenesis using a standard single-primer PCR-based protocol. The threonine to alanine mutant for Venus-cp172Venus FLARE CKAR, as well as the chromophore-dead variant of Venus-cp172Venus FLARE AKAR, was created using Gibson assembly, amplifying the appropriate fragment with a primer containing the desired mutation.

The YFP MLCK FLARE sensor was created by replacing the CFP portion of an existing two color sensor (*Isotani et al., 2004*; *Geguchadze et al., 2004*) with a PCR amplified mVenus fragment flanked by XhoI and AgeI restriction sites. Moreover, the coding sequence for the CFP D1 ER sequence was manufactured by Genewiz. The sensor consists of two oxmCer3 proteins (https://www.ncbi.nlm.nih.gov/pubmed/26158227) flanking the D1 ER calcium sensing domains (*Palmer et al., 2004*) and a C-terminal KDEL ER retention sequence.

## Cell culture and transfection

HEK293T and HeLa cells cells were maintained using Dulbecco's Modified Eagle's Medium (DMEM) supplemented with 10% fetal bovine serum (FBS) and 1% penicillin/streptomycin. Cells were seeded onto a 35-mm glass-bottom imaging dish and incubated at 37°C with 5% ambient carbon dioxide. HEK293T, HeLa, MIN6, and REF-52 cell lines were maintained separately from other cells and were screened regularly to confirm the absence of mycoplasma contamination using Hoechst staining. As the origin of the cells was not central to the nature of these experiments, we did not further validate the identity of the cell lines. Cells were transfected using Lipofectamine 2000 (Invitrogen), Polyjet (SignaGen), or calcium phosphate and incubated for 12–48 hr before imaging. The growth media was removed immediately before imaging, and the cells were washed two or more times with Hanks Balanced Salt Solution (HBSS) buffer with glucose at room temperature. The cells were imaged in HBSS buffer with glucose at either room temperature or 37°C.

## Fluorescence polarization microscopy

Fluorescence anisotropy reporter co-imaging is described in more detail in Bio-protocol (*Ross et al., 2019*). Widefield images were collected using an Zeiss AxioObserver equipped for fluorescence polarization microscopy, using one of two setups. In the first setup, a wire grid polarizer (Meadowlark Optics) was placed in the excitation pathway between the LED illuminators and reflector turret containing filter cubes specific for CFP (Zeiss), YFP (Zeiss), and mCherry (Semrock). Images were generally collected using a 20 × 0.75 NA objective lens. Polarizations parallel and perpendicular to the excitation polarizations were separated using Optical Insights Dual-View using their polarization splitting module. Both images were simultaneously collected in a single image collected by a water-cooled Orca-R2 (Hamamatsu). In the second setup, a polarizer (Chroma) was placed in the excitation pathway between the xenon arclamp and the excitation filters. Images were collected using a 20 × 0.45 NA objective lens. Polarizations were separated using an Opto-Split II LS image splitter, with two wire grid polarizers (Meadowlark) oriented parallel and perpendicular to the excitation polarizer. Images of both polarizations were collected using a Hamamatsu Flash 4.0 sCMOS camera. Two-photon imaging was performed using a Zeiss 7 MP with GaAsP non-descanned detectors housed at the University of Maryland School of Medicine confocal facility. Coherent Chameleon and OPO lasers were used for excitation. Fluorescence was filtered using an ET680 short pass filter for two-photon microscopy (Chroma) prior to separating polarizations with a one inch broadband polarizing beamsplitter cube (Thorlabs) mounted using a custom 3D printed cube. Images were collected using a 10×, 0.3 NA Plan-apochromat objective lens. In vivo imaging was performed on C57Bl/6 mice under isoflurane anesthesia.

## Image analysis

Image analysis was performed using Fiji (ImageJ) open-source image processing software. Polarization images were cropped and aligned using either the Zeiss Axiovision software or Fiji's built-in StackReg registration plugin. In Fiji, regions of interest (ROIs) were drawn around each cell, as well as one in the background. ROI intensities were background subtracted in each channel to estimate fluorescence emission intensity, and anisotropy was calculated as described by *Lakowicz (2010)*. Anisotropies were calculated using the conventional equation (*Geguchadze et al., 2004*):

$$r = \frac{P - gS}{P + 2gS}$$

where $g$ is the correction factor that accounts for differences in polarization transmission efficiencies within the instrument. The g-factor was calculated using an isotropic fluorescein solution as described by *Piston and Rizzo (2008)*. Anisotropy was calculated by subtracting the anisotropy at each time point by the anisotropy at the time point right before drug addition. The magnitude of the anisotropy changes were calculated by taking the difference between the average anisotropy when the signal peaked or plateaued and the average anisotropy of the baseline time points before drug was added.

## Protein purification

Purification of the FLARE-Cameleon sensors was done using the BL21-RIL Codon Plus strain of *E. coli*, which were transformed with the construct cloned in the pRSET-B vector, with a Poly-His tag in the header sequence to allow for metal ion binding. The cells were grown to an OD of 0.2, when expression was induced with IPTG and allowed to grow overnight. The cells were then pelleted, frozen, resuspended, and lysed by sonication. Protein purification was performed using column chromatography with Ni-NTA resin. Fractions were collected and analyzed using SDS-PAGE; fractions showing sufficiently pure protein product were pooled.

## In vitro calcium calibration

In vitro calibration of the FLARE-Cameleon sensor was performed using fluorescence anisotropy spectroscopy in solutions with varying concentrations of free calcium in a temperature-controlled environment. These solutions were made by titrating known concentrations of free EGTA and calcium-saturated EGTA at pH 7.1 (Calcium Concentration Kit #1–Thermo Scientific). The fluorescence anisotropy was measured using a Photon Technology International QuantaMaster spectrofluorometer equipped with a Xenon flash lamp, fluorescence polarizers, and a Peltier cuvette holder for temperature control. Anisotropies were calculated using integrated intensities of S- and P-polarized emission spectra. The correction factor G was measured by measuring the P- and S-polarized emission spectra of fluorescein, which is assumed to be isotropic, and taking the ratio of their measured integrated fluorescence intensities. To determine the dissociation constant ($K_d$) and Hill coefficient (n), the anisotropy vs. calcium concentration data were fit to the following equation:

$$r = r_{min} + (r_{max} - r_{min}) * (\frac{[Ca^{2+}]}{K_d + [Ca^{2+}]})^n$$

## Acknowledgements

We would like to acknowledge Sohum Mehta for his help in creation of graphics as well as critical reading of the manuscript. This work was supported by: NIH R35 CA197622, R01DK073368, and R01MH111516 (to JZ), as well as by R01DK077140, R01HL122827, and R01MH111527 (to MAR).

## Additional information

### Funding

| Funder | Grant reference number | Author |
| --- | --- | --- |
| National Institutes of Health | R35 CA197622 | Jin Zhang |
| National Institutes of Health | R01 DK073368 | Jin Zhang |
| National Institutes of Health | R01 MH111516 | Jin Zhang |
| National Institutes of Health | R01 DK077140 | Megan A Rizzo |
| National Institutes of Health | R01 HL122827 | Megan A Rizzo |
| National Institutes of Health | R01 MH111527 | Megan A Rizzo |

The funders had no role in study design, data collection and interpretation, or the decision to submit the work for publication.

### Author contributions

Brian L Ross, Conceptualization, Data curation, Formal analysis, Validation, Investigation, Methodology, Writing—original draft, Writing—review and editing; Brian Tenner, Data curation, Formal analysis, Validation, Investigation; Michele L Markwardt, Resources, Data curation, Validation, Investigation; Adam Zviman, Guoli Shi, Jaclyn P Kerr, Joseph R Mauban, Investigation; Nicole E Snell, Jennifer J McFarland, Formal analysis, Investigation, Visualization; Christopher W Ward, Resources, Supervision, Investigation, Methodology, Project administration; Megan A Rizzo, Conceptualization, Resources, Data curation, Formal analysis, Supervision, Funding acquisition, Validation, Investigation, Visualization, Methodology, Writing—original draft, Project administration,

Writing—review and editing; Jin Zhang, Conceptualization, Resources, Supervision, Funding acquisition, Methodology, Project administration, Writing—review and editing

### Author ORCIDs
Brian L Ross  http://orcid.org/0000-0001-9020-627X
Megan A Rizzo  https://orcid.org/0000-0001-6528-7768
Jin Zhang  http://orcid.org/0000-0001-7145-7823

### Ethics
Animal experimentation: All work involving mice was performed in accordance and recommendations of the NIH's Guide for the Care and Use of Laboratory Animals. Work was performed under protocols approved by the University of Maryland, Baltimore's Institutional Animal Care and Use Committee (protocol # 1213012). Procedures were performed under isoflurane anesthesia to minimize suffering.

### Decision letter and Author response
Decision letter https://doi.org/10.7554/eLife.35458.sa1
Author response https://doi.org/10.7554/eLife.35458.sa2

## Additional files
### Supplementary files
• Transparent reporting form

### Data availability
Source data have been provided for Figures 1 to 4.

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
