## [Decision Letter]

[Editors’ note: a previous version of this study was rejected after peer review, but the authors submitted for reconsideration. The first decision letter after peer review is shown below.]

Thank you for submitting your work entitled "FLAREs: Single-color, ratiometric biosensors for detecting signaling activities" for consideration by *eLife*. Your article has been reviewed by two peer reviewers, and the evaluation has been overseen by a Reviewing Editor and a Senior Editor.

Our decision has been reached after consultation between the reviewers. Based on these discussions and the individual reviews below, we regret to inform you that your work will not be considered further for publication in *eLife*.

- The concept is novel and the work is potentially a strong candidate for *eLife*. Observation of three different signaling events in parallel is important. However, a reviewer mentioned during the post-review discussion that three reporters in parallel is not by itself new.

- Testing was only done under signal optimal conditions. Example seems cherrypicked.

- Unclear how "usable" the technology will be. A more thorough characterization and comparison to current technology is needed.

- The MAPK sensor clearly should behave differently. The other sensors are also not well characterized.

- Cell population behaviors can be shown (e.g. multiple cell trajectories, average, stdev,.…). This would allow to get from anecdotal results to a real understanding of the robustness of the cell responses and biosensors.

The reviewers were generally positive about the approach but suggest a number of new experiments to validate the method and demonstrate the usability, and to characterize the reporters in more depth. Although I suspect not all the experiments are necessary to address their comments adequately.*eLife*. If a new manuscript is submitted that addresses the point in the future, we will be happy to consult the same reviewers.

Reviewer #1:

The authors re-engineer a number of classic FRET-based biosensors (PKA, ERK, PKC, MLCK, calcium, cAMP) biosensors using single color FPs that perform homo FRET, and that can be measured using fluorescence anisotropy measurements. Since each biosensor is based on only one color, this allows for multiplexed biosensor measurements in single living cells. It also allows to perform biosensor measurements in response to optogenetic activation. Finally, the authors perform some in vivo fluorescence anisotropy measurements using 2-photon microscopy.

The data somehow implies that a combination of mVenus and a cp mutant of Venus cp172-mVenus is the best pair for performing such anisoptropy measurements. It looks like that other pairs involving other color FPs perform much poorly. The authors propose that this can be a generalizable method to perform multiplexed imaging of multiple signaling activities in single living cells. While I like the concept, I think that the method might not be so easy to implement than as proposed. Multiplexed imaging has been performed using experiments that induce very robust signaling states. The poorest performing FP combination has been grafted on a calcium sensor that is known to be one of the most robust signaling modules available. The question is therefore if the approach has the full potential that the authors propose. This detracts my enthusiasm for the manuscript, and I do not think it is publishable in the present form.

Additional comments:

1) Throughout the paper, the authors analyse multiple cells for each respective experiment. Could the authors please show a representation of all of these curves with median, stdev, and accompanying statistics. That would really help the reader to get a sense of the robustness of the biosensor responses.

2) It would be helpful if the authors would calculate the SNR for all the biosensors, as they have performed for their AKAR-FLARE sensor. With 1. This would again give an idea about the robustness of the sensor.

3) The FLARE-EKAR EV sensor in Figure 2A displays sustained ERK activity in response to EGF stimulation, and is then switched off using a MEK inhibitor. This is contrary to another EKAR variant, EKAR2G, that displays a very transient ERK activity in the same HEK293T cells (PMID:23882122). Importantly, this is backed up with a phosphoERK western blot in this paper. What happens with this FLARE EKAR-EV version. Do the new FPs somehow influence the biosensor response, or is this due to the specific HEK293T cells/ experimental conditions used by the authors?

4) Figure 4A: from 9 cells, the authors imply that there are two dynamic signaling states in that experiment. I think that if they want to make a statement about something like this, they need to acquire much more cells, and avoid showing anecdotal data.

5) I think that one of the weaknesses of the paper is that they remain elusive about the microscopy setup, and the image analysis of the datasets. Obviously, all this information is published elsewhere. But this being a technology resource paper, I think that it would really make sense to discuss this in detail both of these aspects to make the reader more familiar with this imaging technology.

Reviewer #2:

The study of molecular dynamics in single cells has revealed unexpected complexity, only tractable by multiplexing molecular biosensors. Ross, et al. explore the use of fluorescence anisotropy to measure conformational changes in genetically encoded biosensors using a single color per sensor. The authors successfully use ratiometric measurements of polarized fluorescence to enable quantification of previously described FRET reporters. This is a very exciting idea as it could substantially increase multiplexing capabilities in single cells. The authors validate the use of FLAREs in a variety of individual sensors, multiplexed with optogenetic tools and in vivo. Although I find the concept and validation of great interest, the characterization of intramolecular fluorescence anisotropy and comparison to current methods is limited.

1) The authors should explore the use of fluorescence anisotropy to measure subcellular changes in activity. One of the advantages of FRET sensors is the ability to measure spatially restricted changes in protein activity (within cell compartments), however, the paper does not address whether local activities can be monitored also by fluorescence anisotropy. In fact, given the single color nature of FLAREs, single cell comparisons of FRET vs FLARE should be possible and helpful to define how these technologies compare quantitatively.

2) The novelty of this study relies on the use of intramolecular fluorescence anisotropy to measure conformational changes. However, the relationship between intramolecular distance (or orientation) and anisotropy is not explored. These experiments could be done by using linkers of different lengths or known proteins with different structures. In my opinion, a more thorough characterization of the parameters that affect intramolecular anisotropy is needed to understand the full spectrum of limitations and possibilities of this method.

3) The authors find that a circularly permutated Venus together with wt Venus is better at changing anisotropy, but then, the comparison to other FPs is done without any circular permutation. It is not clear, from the manuscript, what are the reasons for such comparison. If circular permutation affects anisotropy, other FPs should also be permuted.

4) Most experiments to test the reporting abilities of the sensor are done under chemical perturbation with chemicals that create non physiological states (i.e. Fsk, Ionomycin, PMA). To determine whether the reporting range of the sensor is appropriate for "real" conditions, a dose-response under physiological stimulation compared to other sensors should be done.

5) The choice of fluorescent proteins in the dual sensor experiment provided in Figure 4B is unfortunate. Venus and Cerulean3 are a FRET pair. This could be confusing the interpretation of the results. Using the Cherry version in one of the two sensors would be a more appropriate choice.

Overall, this study offers a novel strategy to enhance multiplexing capabilities when measuring single cell dynamics. Although the authors show multiple proof-of-principle experiments, the quantitative description of the parameters that affect intramolecular anisotropy and how it compares to FRET is poorly explored. Thus, I would encourage resubmission when these issues are addressed.

[Editors’ note: what now follows is the decision letter after the authors submitted for further consideration.]

Thank you for submitting your article "FLAREs: Single-color, ratiometric biosensors for detecting signaling activities" for consideration by *eLife*. Your article has been reviewed by two peer reviewers, and the evaluation has been overseen by a Reviewing Editor and Jonathan Cooper as the Senior Editor. The reviewers have opted to remain anonymous.

The reviewers have discussed the reviews with one another and the Reviewing Editor has drafted this decision to help you prepare a revised submission.

Summary:

The authors re-engineered a number of classic FRET-based biosensors (PKA,ERK, PKC, MLCK, calcium, cAMP) using single color FP pairs that can perform homo FRET, which can be measured using fluorescence anisotropy measurements. Since each biosensor is based on only one color, this allows for multiplexed biosensor measurements in single living cells. It also allows them to perform biosensor measurements in combination with optogenetic activation systems. Finally, the authors perform some duo-color in vivo fluorescence anisotropy measurements using 2-photon microscopy. The data convincingly shows that (cp)mVenus and mCherry anistropy sensors can be multiplexed in complex cell types and in vivo measurements. The data also shows that the (cp) mCer-sensors performs a lot worse in comparison. Overall, the concept and experimental validation reported here are of great interest for the research community and the revision improved the quality of the work significantly. However, there are still significant concerns some of which may require some additional experiments to address.

Essential revisions:

1) The authors stress that their method provides a generalizable method for (triple) multiplex imaging, and state that "The fact that FLAREs only occupy a single color channel and are highly generalizable for different biosensors, as well as color variants, highlights their utility for multiplexed imaging applications".

With such a strong statement:

The authors should really show that the mCer3 based anisotropy measurements are usable in multiplex experiments beyond highly sensitive calcium sensor modules and / or under physiological conditions (e.g. histamine-induced response on FLARE-mCer3 probe).

2) The authors added (Figure 1—figure supplement 1) a convincing comparison of different linkers in mVenus-mVenus anisotropy measurements. Is this a general principle that can be applied to optimize anisotropy based biosensors? E.g. is this similar for mCherry/mCerulean based anisotropy sensors? This should at least be discussed (or better shown) to provide the readers with a basis to start utilising this technique. Similarly, it would greatly help the reader audience if the authors can discuss/speculate on the reason for different performance of the mVenus-pair over the other fluorescent protein pairs in anisotropy measurements.

3) One of the advantages of FRET sensors is the ability to measure spatially restricted changes in protein activity (within cell compartments), however, the paper does not address whether local activities can be resolved also by fluorescence anisotropy. In Figure 1—figure supplement 5 authors target the reporters to subcellular compartments and show changes in anisotropy, however, I think the authors should determine whether a local activity can be resolved using a non localized sensor. For instance Matsuda et al. (PMCID: PMC3226481) show that PKC in response to TPA is activated at the membrane edge. The authors should test the spatial resolution power of anisotropic probes using their FLARE CKAR.

4) All the experiments shown in this study have been done using transfection or electroporation which results in major overexpression of the biosensor. The authors should determine whether expression level determines the reporting ability of these sensors.

5) Although it may seem obvious I think the authors should determine whether the anisotropy change depends on the fluorescent proteins being in close proximity or just the conformational change itself. Mutating one chromophore could easily address this question.

6) Statistical methods and data reporting:

- Generally: Why are the average curves (including the variance measurements) in the supplemental figures? It would be much more informative and convincing for readers to show these in the main figures instead of the current "representative example curves".

- Unclear from which timepoints (or pooled timepoints?) the data in the boxplots in Figure 1B, C, 2A, B, C is calculated/compared.

- Unclear whether mean or median is represented in the boxplots in:

Figure 1B, C

Figure 2A, B, C

Figure 1—figure supplement 3 (also which variance measurement is used here?)

Figure 3—figure supplement 1B

Figure 3—figure supplement 3

Figure 4E (which variance measurement is used here, how is statistical significance calculated? -> descriptions missing in legend/main text).

- No (vehicle) controls in Figure 2B, C e.g. MLCK sensors can be activated by mechanical stress on cells from adding the experimental agents (it is also unclear from the "Materials and methods" section in what chamber cells were imaged, and how agents were added to the cells during experiments).

---

## [Author Response]

[Editors’ note: the author responses to the first round of peer review follow.]

We would like to express our gratitude to the editor and the reviewers for their time and effort in considering our original submission and providing suggestions for improving the manuscript. The new version incorporates a number of changes, including additional experimental data and text, in accordance with the recommendations of the reviewers. We believe that we have fully addressed the concerns raised by the reviewers.

- The concept is novel and the work is potentially a strong candidate for eLife. Observation of three different signaling events in parallel is important. However, a reviewer mentioned during the post-review discussion that three reporters in parallel is not by itself new.

We are excited that the editor finds the concept novel and a strong candidate for publication in *eLife*. Monitoring multiple signaling events in parallel is important. Currently, existing co-imaging approaches remain largely confined to monitoring two activities in parallel. Monitoring three or more activities in the same cell is neither routine nor generalizable, and current examples have to be highly tailored to specific applications or highly specialized optical setups.^1,2^ For example, a three-parameter co-imaging study utilized a biosensor for cGMP based on blue fluorescent protein and a nonfluorescent-YFP quenching acceptor, a CY-FRET cAMP biosensor and Fura red, for cGMP, cAMP and Ca^2+^ co-imaging.^3^ We believe that FLAREs offers a generalizable way to monitor multiple activities simultaneously in the same cell, while preserving many of the advantages of ratiometric fluorescent sensors.

- Testing was only done under signal optimal conditions. Example seems cherrypicked.

Several examples of suboptimal conditions were used. For example, cAMP and calcium oscillations in MIN6 cells are sub-maximal conditions where small oscillatory changes in second messenger levels can be monitored using the FLARE probes (Figure 4B, Figure 4—figure supplement 2). Furthermore, we demonstrate the use of FLAREs in vivo in the muscle cells of live mice (Figure 4D, E). In the revised version of the manuscript, we also include an isoproterenol dose response curve for a FLARE AKAR (Figure 1—figure supplement 3), as well as a supplementary figure showing histamine-induced responses of a FLARE calcium probe (Figure 3—figure supplement 1C). Together, these data demonstrate that FLARE probes can detect responses under physiological stimulation conditions.

For co-imaging experiments, we have used different color/biosensor combinations:

mCherry-mCherry FLARE-AKAR, Venus-cp172Venus FLARE-EKAR, and mCerulean3-mCerulean3 FLARE-cameleon in HEK293 cells (Figure 4A); Cerulean3-Cerulean3 FLARE-cameleon and Venus-cp172Venus FLARE-ICUE in MIN6 cells (Figure 4B); and Venus-cp172Venus FLARE-cameleon and mCherry-mCherry FLARE-AKAR for in vivo imaging (Figure 4D, E).

The different biosensor combinations used in these co-imaging experiments, the different stimulation conditions and the successful demonstration of multiplexed in vivo imaging together suggest that the FLARE probes are useful for co-imaging in different contexts.

- Unclear how "usable" the technology will be. A more thorough characterization and comparison to current technology is needed.

In the revised manuscript, we have added an experiment to directly compare a FLARE biosensor with a FRET biosensor, as requested by the reviewer. Specifically, we co-expressed AKAR4, the current-generation FRET-based PKA activity biosensor and mCh-mCh FLARE AKAR in the same cell and co-imaged these two biosensors (Figure 1—figure supplement 4). The data show that FLARE AKAR captures all the basic characteristics of the response of FRET-based AKAR. In addition, we have included an SNR table of all the sensors and added a discussion comparing FLARE probes to the existing technology. We have also included data to demonstrate the utility of targeted versions of the AKAR sensor.

- The MAPK sensor clearly should behave differently. The other sensors are also not well characterized.

There are cell-to-cell variations in terms of the dynamics of the ERK response to EGF. Some cells in the population do indeed show more transient ERK responses (Figure 4—figure supplement 1). These variations may result from variations in the expression levels of the receptor and/or other components, which may also result in differences observed in different sub-lines of HEK293T cells. We included here data from FRET-based EKAR, which shows the same response pattern and heterogeneity of the dynamics of ERK response to EGF. Specifically, on the time-scale of our experiment, some cells show a nearly-sustained response, while other show a more transient response. In the revised manuscript, we have also included more detailed characterization of FLARE sensors as the reviewers suggested (please see below).

- Cell population behaviors can be shown (e.g. multiple cell trajectories, average, stdev,.…). This would allow to get from anecdotal results to a real understanding of the robustness of the cell responses and biosensors.

In the revised draft, we have included individual cell traces, as well as average curves with standard error of the mean for all the data shown in Figures 1-4. We have also included an SNR table to compare the performance of each sensor (Table 1).

The reviewers were generally positive about the approach but suggest a number of new experiments to validate the method and demonstrate the usability, and to characterize the reporters in more depth. Although I suspect not all the experiments are necessary to address their comments adequately. If a new manuscript is submitted that addresses the point in the future, we will be happy to consult the same reviewers.Reviewer #1:The authors re-engineer a number of classic FRET-based biosensors (PKA, ERK, PKC, MLCK, calcium, cAMP) biosensors using single color FPs that perform homo FRET, and that can be measured using fluorescence anisotropy measurements. Since each biosensor is based on only one color, this allows for multiplexed biosensor measurements in single living cells. It also allows to perform biosensor measurements in response to optogenetic activation. Finally, the authors perform some in vivo fluorescence anisotropy measurements using 2-photon microscopy.The data somehow implies that a combination of mVenus and a cp mutant of Venus cp172-mVenus is the best pair for performing such anisoptropy measurements. It looks like that other pairs involving other color FPs perform much poorly. The authors propose that this can be a generalizable method to perform multiplexed imaging of multiple signaling activities in single living cells. While I like the concept, I think that the method might not be so easy to implement than as proposed. Multiplexed imaging has been performed using experiments that induce very robust signaling states. The poorest performing FP combination has been grafted on a calcium sensor that is known to be one of the most robust signaling modules available. The question is therefore if the approach has the full potential that the authors propose. This detracts my enthusiasm for the manuscript, and I do not think it is publishable in the present form.

We would like to thank the reviewer for the thoughtful comments.

The SNRs for the FLARE biosensors were determined to be 32 for Venus-cpVenus FLARE AKAR, 10 for Cherry-Cherry FLARE AKAR, and 3 for Cerulean-Cerulean FLARE AKAR. In fact, all of the current FLARE probes show an SNR ≥3, with many above 15 (Table 1). While there is definitely room for improvement, we believe that this performance is comparable with first generation FRET sensors. We have now added a brief discussion about this point.

In terms of non-optimal conditions, several examples of suboptimal conditions were used. For example, cAMP and calcium oscillations in MIN6 cells are sub-maximal conditions where small oscillatory changes in second messenger levels can be consistently detected using the FLARE probes (Figure 4B, Figure 4—figure supplement 2). Furthermore, we demonstrated multiplexed in vivo imaging using skeletal muscle of the foot of a live mouse (Figure 4D, E). It is well known that in vivo imaging presents the most challenging test of biosensor performance as reduced dynamic range is often observed in these settings. We have shown that FLARE biosensors can provide a multiplexed readout of signaling activities in vivo. In the revised version of the manuscript, we also include an isoproterenol dose response curve for a FLARE AKAR (Figure 1—figure supplement 3), as well as a supplementary figure showing histamine-induced responses of a FLARE calcium probe (Figure 3—figure supplement 1C). Together, these data demonstrate that FLARE probes can detect responses under physiological stimulation conditions.

For these co-imaging experiments, we have used different color/biosensor combinations:

mCherry-mCherry FLARE-AKAR, Venus-cp172Venus FLARE-EKAR, and mCerulean3-mCerulean3 FLARE-cameleon in HEK293 cells; Cerulean3-Cerulean3 FLARE-cameleon and Venus-cp172Venus FLARE-ICUE in MIN6 cells; and Venus-cp172Venus FLARE-cameleon and mCherry-mCherry FLARE-AKAR for in vivo imaging.

The different biosensor combinations used in these co-imaging experiments, the different stimulation conditions and the successful demonstration of multiplexed in vivo imaging together suggest that the FLARE probes are useful for co-imaging in different contexts.

Additional comments:1) Throughout the paper, the authors analyse multiple cells for each respective experiment. Could the authors please show a representation of all of these curves with median, stdev, and accompanying statistics. That would really help the reader to get a sense of the robustness of the biosensor responses.

We have now added multiple cell traces as well as the average curve with standard error, as well as other relevant statistics to the supplementary information, in addition to the representative trace (Figure 1—figure supplement 2, Figure 2—figure supplement 1, Figure 2—figure supplement 3, Figure 2—figure supplement 4, Figure 3—figure supplement 1, Figure 3—figure supplement 4, Figure 4—figure supplement 1, and Figure 4—figure supplement 2).

2) It would be helpful if the authors would calculate the SNR for all the biosensors, as they have performed for their AKAR-FLARE sensor. With 1. This would again give an idea about the robustness of the sensor.

We have now added a table of the SNR for all the sensors presented in the manuscript (Table 1).

3) The FLARE-EKAR EV sensor in Figure 2A displays sustained ERK activity in response to EGF stimulation, and is then switched off using a MEK inhibitor. This is contrary to another EKAR variant, EKAR2G, that displays a very transient ERK activity in the same HEK293T cells (PMID:23882122). Importantly, this is backed up with a phosphoERK western blot in this paper. What happens with this FLARE EKAR-EV version. Do the new FPs somehow influence the biosensor response, or is this due to the specific HEK293T cells/ experimental conditions used by the authors?

There are cell-to-cell variations in terms of the dynamics of the ERK response to EGF. Some cells in the population do indeed show transient more transient ERK responses (Figure 4—figure supplement 1). These variations may result from variations in the expression levels of the receptor and/or other components, which may also result in differences observed in different sub-lines of HEK293T cells. We included here data from FRET-based EKAR, which shows the same response pattern and heterogeneity of the dynamics of ERK response to EGF. Specifically, on the time-scale of our experiment, some cells show a nearly-sustained response, while other show a more transient response.

**Author response image 1. respfig1:** Single cell traces of EKAR2.3-EV, in HEK cells.EKAR2.3-EV with a nuclear exclusion sequence was expressed in HEK cells, and cells were stimulated with 100ng/mL epidermal growth factor (EGF) at t=0.

4) Figure 4A: from 9 cells, the authors imply that there are two dynamic signaling states in that experiment. I think that if they want to make a statement about something like this, they need to acquire much more cells, and avoid showing anecdotal data.

Since this behavior was observed in a small sub-population of cells, we have now removed the specific note.

5) I think that one of the weaknesses of the paper is that they remain elusive about the microscopy setup, and the image analysis of the datasets. Obviously, all this information is published elsewhere. But this being a technology resource paper, I think that it would really make sense to discuss this in detail both of these aspects to make the reader more familiar with this imaging technology.

We have included additional details concerning microscopy setup and image analysis, including the identification and positioning of the excitation polarizer, method for measuring and calculating the corrective g-factor, and the anisotropy equation.

Reviewer #2:The study of molecular dynamics in single cells has revealed unexpected complexity, only tractable by multiplexing molecular biosensors. Ross, et al. explore the use of fluorescence anisotropy to measure conformational changes in genetically encoded biosensors using a single color per sensor. The authors successfully use ratiometric measurements of polarized fluorescence to enable quantification of previously described FRET reporters. This is a very exciting idea as it could substantially increase multiplexing capabilities in single cells. The authors validate the use of FLAREs in a variety of individual sensors, multiplexed with optogenetic tools and in vivo. Although I find the concept and validation of great interest, the characterization of intramolecular fluorescence anisotropy and comparison to current methods is limited.

We would like to thank the reviewer for the thoughtful comments. We have now added the characterization of intramolecular fluorescence anisotropy and comparison to current methods.

1) The authors should explore the use of fluorescence anisotropy to measure subcellular changes in activity. One of the advantages of FRET sensors is the ability to measure spatially restricted changes in protein activity (within cell compartments), however, the paper does not address whether local activities can be monitored also by fluorescence anisotropy. In fact, given the single color nature of FLAREs, single cell comparisons of FRET vs FLARE should be possible and helpful to define how these technologies compare quantitatively.

In the revised submission, we have now included examples of FLARE sensors targeted to different subcellular compartments. Specifically we have demonstrated the ability to monitor local kinase activities at the plasma membrane and the outer mitochondrial membrane (Figure 1—figure supplement 5).

As the reviewer suggested, we have also added an experiment to directly compare a FLARE biosensor with a FRET biosensor. Specifically, we co-expressed AKAR4, the current-generation FRET-based PKA activity biosensor, and mCh-mCh FLARE AKAR in the same cell and co-imaged the two probes (Figure 1—figure supplement 4). The data show that FLARE AKAR captures all the basic characteristics of the response of FRET-based AKAR.

2) The novelty of this study relies on the use of intramolecular fluorescence anisotropy to measure conformational changes. However, the relationship between intramolecular distance (or orientation) and anisotropy is not explored. These experiments could be done by using linkers of different lengths or known proteins with different structures. In my opinion, a more thorough characterization of the parameters that affect intramolecular anisotropy is needed to understand the full spectrum of limitations and possibilities of this method.

The anisotropy measurement in these experiments is a way of quantifying homoFRET. The relationship between intramolecular distance and anisotropy can therefore be predicted based on the theory of FRET. To validate this, we have now included the requested experiment (Figure 1—figure supplement 1) that shows the relationship between linker length and fluorescence depolarization using mVenus. A single mVenus emits light with the highest measured anisotropy, whereas two tandem mVenus:mVenus fusion constructs both displayed reduced anisotropy due to FRET. The shorter linker construct exhibited a greater reduction compared with the longer linker version due to the higher homo-FRET efficiency.

3) The authors find that a circularly permutated Venus together with wt Venus is better at changing anisotropy, but then, the comparison to other FPs is done without any circular permutation. It is not clear, from the manuscript, what are the reasons for such comparison. If circular permutation affects anisotropy, other FPs should also be permuted.

We have now included data on cp173-mCerulean in the acceptor position for the FLARE AKAR (Figure 1C).

4) Most experiments to test the reporting abilities of the sensor are done under chemical perturbation with chemicals that create non physiological states (i.e. Fsk, Ionomycin, PMA). To determine whether the reporting range of the sensor is appropriate for "real" conditions, a dose-response under physiological stimulation compared to other sensors should be done.

We have shown that EGF-induced ERK activity can be detected by FLARE-EKAR. In the revised draft, we have now included a dose-response curve of FLARE AKAR using the β-adrenergic receptor agonist isoproterenol (Figure 1—figure supplement 3). Furthermore, we have also included calcium measurements from stimulation of Gq-coupled receptors with histamine (Figure 3—figure supplement 1C). These experiments demonstrate the capability of the FLARE sensors to report physiologically relevant responses.

5) The choice of fluorescent proteins in the dual sensor experiment provided in Figure 4B is unfortunate. Venus and Cerulean3 are a FRET pair. This could be confusing the interpretation of the results. Using the Cherry version in one of the two sensors would be a more appropriate choice.

While we agree that the broad spectral properties of FPs allow for many potential FRET combinations, including mCerulean3:Venus, mVenus:mCherry, and even EGFP:mCherry, each of these combinations has been widely used for non-FRET two-color experiments.^4,5^ The imaging conditions we use prevent bleedthrough of fluorescence into non-specific channels, permitting clear and complete separation of mCerulean3 and mVenus fluorescence (Author response image 2).

**Author response image 2. respfig2:** a). Agarose beads were separately labeled with recombinant mCerulean3 or mVenus. Imaging was performed using collection conditions specific for mCerulean3 or mVenus. b). The fluorescence intensity above background was measure for CFP and YFP labeled beads in each channel (n=10 beads per fluorescent protein; ****, p < 0.0001 by t-test.

Citations:

1) Depry C, Mehta S, Zhang J. Multiplexed visualization of dynamic signaling networks using genetically encoded fluorescent protein-based biosensors. Pflugers Arch. 2013;465(3):373-81.

2) Carlson HJ, Campbell RE. Genetically encoded FRET-based biosensors for multiparameter fluorescence imaging. Curr Opin Biotechnol. 2009;20(1):19-27.

3) Niino Y, Hotta K, Oka K. Blue fluorescent cGMP sensor for multiparameter fluorescence imaging. PLoS ONE. 2010;5(2):e9164.

4) Markwardt ML, Nkobena A, Ding SY, Rizzo MA. Association with nitric oxide synthase on insulin secretory granules regulates glucokinase protein levels. Mol Endocrinol. 2012;26(9):1617-29.

5) Gaal T, Bratton BP, Sanchez-vazquez P, et al. Colocalization of distant chromosomal loci in space in E. coli: a bacterial nucleolus. Genes Dev. 2016;30(20):2272-2285.

Overall, this study offers a novel strategy to enhance multiplexing capabilities when measuring single cell dynamics. Although the authors show multiple proof-of-principle experiments, the quantitative description of the parameters that affect intramolecular anisotropy and how it compares to FRET is poorly explored. Thus, I would encourage resubmission when these issues are addressed.

[Editors' note: the author responses to the re-review follow.]

Essential revisions:1) The authors stress that their method provides a generalizable method for (triple) multiplex imaging, and state that "The fact that FLAREs only occupy a single color channel and are highly generalizable for different biosensors, as well as color variants, highlights their utility for multiplexed imaging applications".With such a strong statement:The authors should really show that the mCer3 based anisotropy measurements are usable in multiplex experiments beyond highly sensitive calcium sensor modules and / or under physiological conditions (e.g. histamine-induced response on FLARE-mCer3 probe).

We demonstrated that the mCer3-mCer3 FLARE cameleon sensor could be used to monitor calcium oscillations in Min6 cells in co-imaging experiments with mVenus-cp172Venus FLARE ICUE. The calcium responses are submaximal in this context. In addition, in this revision we included two additional multiplexed co-imaging experiments to demonstrate that the mCer3 FLARE sensors could detect physiologically relevant responses. First, we performed a co-imaging experiment of mCer3-mCer3 FLARE Cameleon, mCh-mCh FLARE AKAR, and Venus-cp172Venus FLARE EKAR in HeLa cells, with Fsk/IBMX, EGF, and histamine stimulation (Figure 4—figure supplement 2). In the HeLa cells, calcium transients were observed both after EGF stimulation and after histamine stimulation. Furthermore, we also co-imaged mCer3-cp173Cer3 FLARE AKAR and Venus-cp172 Venus cameleon with 100nM isoproterenol, a physiological stimulant, and thapsigargin (Figure 4—figure supplement 3). Changes in anisotropy were observed in both channels upon the appropriate stimulation.

2) The authors added (Figure 1—figure supplement 1) a convincing comparison of different linkers in mVenus-mVenus anisotropy measurements. Is this a general principle that can be applied to optimize anisotropy based biosensors? E.g. is this similar for mCherry/mCerulean based anisotropy sensors? This should at least be discussed (or better shown) to provide the readers with a basis to start utilising this technique. Similarly, it would greatly help the reader audience if the authors can discuss/speculate on the reason for different performance of the mVenus-pair over the other fluorescent protein pairs in anisotropy measurements.

We performed the experiment comparing the anisotropy of monomeric FP, and dimers separated by linkers of two different lengths for mCherry and mCerulean3, in addition to mVenus (Figure 1—figure supplement 1). A similar trend was observed. Monomeric FPs had the highest anisotropy, followed by dimers separated by a long linker, with dimers separated by a short linker having the lowest measured anisotropy. This observation is consistent with our understanding of homo-FRET, with increased FRET efficiency corresponding to a decrease in anisotropy of the emitted light.

For FLARE sensors, like FRET sensors, it is difficult to know a priori which FPs or FP combinations will yield the greatest dynamic range. However, we have observed for a number of different sensors that the mVenus based ones tend to perform better. We speculate that this is because mVenus is both a good FRET donor and a FRET acceptor because of its excellent quantum yield and extinction coefficient.

3) One of the advantages of FRET sensors is the ability to measure spatially restricted changes in protein activity (within cell compartments), however, the paper does not address whether local activities can be resolved also by fluorescence anisotropy. In Figure 1—figure supplement 5 authors target the reporters to subcellular compartments and show changes in anisotropy, however, I think the authors should determine whether a local activity can be resolved using a non localized sensor. For instance Matsuda et al. (PMCID: PMC3226481) show that PKC in response to TPA is activated at the membrane edge. The authors should test the spatial resolution power of anisotropic probes using their FLARE CKAR.

The PKC sensor in Matsudaet al. contains the C1 domain of PKCβ at N-terminus and translocates from the cytosol to the plasma membrane upon activation, which is conveniently used to enhance the detection of the membrane activation event.

To demonstrate the ability to monitor localized activity using an untargeted FLARE sensor, we used mVenus-cp172Venus FLARE AKAR in HeLa cells and stimulated the cells with Fsk/IBMX. We observed clear differences, both in terms of the kinetics and the magnitude of the signals (Figure 1—figure supplement 8), demonstrating that localized signals can be distinguished using non-localized probes.

4) All the experiments shown in this study have been done using transfection or electroporation which results in major overexpression of the biosensor. The authors should determine whether expression level determines the reporting ability of these sensors.

A scatterplot of anisotropy change vs. intensity (expression level) has been added for the Venus-cp172Venus FLARE AKAR and mCherry-mCherry FLARE AKAR as examples (Figure 1—figure supplement 3). In general, the expression level does not significantly impact the reporting ability of these sensors.

5) Although it may seem obvious I think the authors should determine whether the anisotropy change depends on the fluorescent proteins being in close proximity or just the conformational change itself. Mutating one chromophore could easily address this question.

We mutated the chromophore of the C-terminal cp172Venus in Venus-cp172Venus FLARE AKAR from GYG to GGG, and performed anisotropy imaging of this sensor in HEK cells with Fsk/IBMX stimulation of PKA (Figure 1—figure supplement 5). This mutation did not completely abolish the response but instead reduced the response amplitude to about one-third of the wildtype biosensor. The anisotropy change was not completely abolished, presumably, because of intermolecular FRET occurring between biosensor molecules. After stimulation of PKA activity, it is possible for a proportion of the FHA1-PKA substrate interactions to occur between two adjacent sensor molecules, resulting in FRET and subsequently a slight change in measured anisotropy.

We think it is unlikely that the anisotropy changes observed in the FLARE sensors are due to a change in the rotational mobility of the molecules. It has been shown that the rotational mobility of FPs occurs on a timescale much longer than the lifetime of the fluorescence. Thus the contribution to a change in rotational mobility due to a change in the shape of the sensor to the dynamic range should be negligible.

6) Statistical methods and data reporting:- Generally: Why are the average curves (including the variance measurements) in the supplemental figures? It would be much more informative and convincing for readers to show these in the main figures instead of the current "representative example curves".

We have replaced the representative curves with average curves with standard error of the mean in the main figures (Figures 1, 2, and 3). However, for the co-imaging experiments we have kept the representative curves in the main figure to emphasize the single-cell nature of this technique (Figure 4).

- Unclear from which timepoints (or pooled timepoints?) the data in the boxplots in Figure 1B, C, 2A, B, C is calculated/compared.

Time points for calculating these responses have been included in each figure legend.

- Unclear whether mean or median is represented in the boxplots in:Figure 1B, CFigure 2A, B, CFigure 1—figure supplement 3 (also which variance measurement is used here?)Figure 3—figure supplement 1BFigure 3—figure supplement 3Figure 4E (which variance measurement is used here, how is statistical significance calculated? -> descriptions missing in legend/main text).

The fact that the mean and standard error are represented in the dot plots has been added to the appropriate figure legends. Descriptions of statistical tests have also been added.

- No (vehicle) controls in Figure 2B, C

For the Venus-cp172Venus FLARE CKAR sensor, a negative control in which the phosphorylatable threonine in the substrate was mutated to an alanine was added (Figure 2B, Figure 2—figure supplement 3B). A vehicle control was added for the FLARE MLCK sensor (Figure 2C).

E.g. MLCK sensors can be activated by mechanical stress on cells from adding the experimental agents (it is also unclear from the "Materials and methods" section in what chamber cells were imaged, and how agents were added to the cells during experiments).

An explanation has been added to the Materials and methods section.